# Vessel-Sparing Lymphadenectomy Should Be Performed in Small Intestine Neuroendocrine Neoplasms

**DOI:** 10.3390/cancers14153610

**Published:** 2022-07-25

**Authors:** Detlef K. Bartsch, Sebastian Windel, Veit Kanngießer, Moritz Jesinghaus, Katharina Holzer, Anja Rinke, Elisabeth Maurer

**Affiliations:** 1Department of Visceral-, Thoracic- and Vascular Surgery, Philipps-University Marburg, Baldingerstraße, 35043 Marburg, Germany; bartsch@med.uni-marburg.de (D.K.B.); sebastian.windel@uk-gm.de (S.W.); kanngies@med.uni-marburg.de (V.K.); katharina.holzer@uk-gm.de (K.H.); 2Institute of Pathology, Philipps-University Marburg, Baldingerstraße, 35043 Marburg, Germany; moritz.jesinghaus@uni-marburg.de; 3Department of Gastroenterology and Endocrinology, Philipps-University Marburg, Baldingerstraße, 35043 Marburg, Germany; sprengea@uni-marburg.de

**Keywords:** small intestine neuroendocrine neoplasms, primary tumor resection, lymphadenectomy, vessel-sparing

## Abstract

**Simple Summary:**

Primary tumor resection with lymphadenectomy in small intestine neuroendocrine neoplasms potentially requires extensive small bowel resections due to central lymph node metastases and mesenteric fibrosis. Retrograde vessel-sparing lymphadenectomy (VS-LA) might be a sufficient method for avoiding local recurrence and for sparing the small bowel at the same time. We retrospectively analyzed clinical, surgical and pathological data of 50 patients with SI-NENs who exclusively underwent small bowel resections; half of them received conventional lymphadenectomy and 25 underwent VS-LA. VS-LA resulted in shorter resected bowel segments (median 40 cm vs. 65 cm, *p* = 0.007) with similar rates of local R0 resections (72% vs. 84%) and number of resected lymph nodes (median 13 vs. 13). Postoperative complications occurred significantly less in the vessel-sparing group. VS-LA should be the preferred surgical method in small bowel resections for SI-NENs.

**Abstract:**

Introduction: The goal of primary tumor resection with lymphadenectomy (PTR) in small intestine neuroendocrine neoplasms (SI-NENs) is to avoid local recurrence while sparing as much of the small bowel as possible, even in the case of extensive mesenteric fibrosis. The results of PTR with retrograde vessel-sparing lymphadenectomy (VS-LA) were compared to those of conventional lymphadenectomy (Con-LA). Methods: Prospectively collected clinical, surgical and pathological data of consecutive patients with SI-NENs who underwent small bowel resections were retrospectively analyzed regarding the resection technique performed. Results: In a 7-year period, 50 of 102 patients with SI-NENs had only small bowel resections; of those, 25 were VS-LA and 25 were Con-LA. Patients with VS-LA had tendentially more advanced diseases with slightly higher rates of abdominal pain, mesenteric shrinkage and more level III lymph node involvement compared to patients with Con-LA. VS-LA, however, resulted in shorter resected bowel segments (median 40 cm vs. 65 cm, *p* = 0.007) with similar rates of local R0 resections (72% vs. 84%) and resected lymph nodes (median 13 vs. 13). Postoperative clinically relevant complications occurred in 1 of 25 (4%) in the VS-LA and in 7 of 25 (28%) patients in the Con-LA group (*p* = 0.02). Three months after surgery, 1 of 25 (4%) patients of the VS-LA group and 10 of 25 (40%) patients in the Con-LA group (*p* = 0.002) complained about abdominal pain. One of eight patients in the VS-LA group and two of thirteen patients in the Con-LA group who had completely resected stage III disease complained about diarrhea (*p* = 0.31). Conclusion: VS-LA seems to be oncologically safe and should be considered in small bowel resections for SI-NENs.

## 1. Introduction

Small intestine neuroendocrine neoplasms (SI-NENs) are the most prevalent small bowel neoplasms [1,2]. SI-NENs are usually small (<20 mm), located in the ileum and occur in multiples in about 30% of cases [3,4,5]. Despite being very small and having low proliferation, SI-NENs have a high tendency to metastasize. At diagnosis, liver metastases are present in approximately 50% of patients and mesenteric lymph node metastases are present in more than 80% of patients [6,7,8,9]. Regional lymph node metastases are often larger than the primary tumor(s) and frequently associated with dense desmoplastic fibrosis, leading to mesenteric shrinkage. This condition can cause acute symptoms such as abdominal pain, bowel obstruction and mesenteric ischemia, which may require emergency intervention in up to 25% of cases [10]. Surgical treatment is the cornerstone of multimodal SI-NEN management and may be curative in cases of complete R0 resection. However, an R0 resection is feasible in only 20–30% of patients because of the usually advanced disease at diagnosis [11,12]. The main rules for SI-NEN surgery are the resection of all small bowel tumors in association with a systematic lymphadenectomy, while preserving as much of the small bowel as possible [11,12]. Short bowel syndrome should definitively be avoided. It has been demonstrated that the length of the resected small bowel is not correlated with the number of resected lymph nodes [13]. Previously, a V-shaped conventional lymphadenectomy (Con-LA), also called the “pizza pie” technique, was the standard procedure. Often, a significant part of the small bowel was resected. Currently, some groups advocate for primary tumor resection with retrograde vessel-sparing lymphadenectomy (VS-LA) with the goals of reducing the length of resected small bowel and of avoiding postoperative diarrhea or even short bowel syndrome by achieving the same oncological radicality [14,15]. In this retrospective study, the results of VS-LA and Con-LA regarding complications, radicality and postoperative functionality were analyzed.

## 2. Material and Methods

### 2.1. Patients

Between April 2014 and December 2021, a total of 102 patients with SI-NENs were subjected to surgery at our institution. Some details of these patients, such as pathological data, metastatic progression, long-term survival and prognostic markers have been previously reported [6,10]. In the present study, only patients operated on after 2014 with small bowel resections who were subjected to primary tumor resection with either vessel-sparing lymphadenectomy (VS-LA) or conventional lymphadenectomy (Con-LA) were analyzed. Patients’ demographics, clinical and pathological data were retrieved from the prospective database of the Marburg ENETS Center of Excellence and retrospectively analyzed. Only patients with a confirmed histopathological diagnosis of SI-NENs made through microscopy and positive immunohistochemical staining of surgical specimens for chromogranin A and synaptophysin were included after informed written consent.

In addition, operating reports were scrutinized by 2 authors (S.W. and D.K.B.) to define the level of lymph node metastases according to a simplified, more practical classification, based on Ohrvall et al. [7], of the presence of mesenteric shrinkage and the type and extent of lymphadenectomy performed. The levels of lymph node metastases were modified as follows: level 1, lymph nodes from the small bowel to the junction of the ileocolic artery and superior mesenteric artery; level 2, lymph nodes from the junction of the ileocolic artery and superior mesenteric artery up to the horizontal part of the duodenum and the lower border of the pancreatic body; and level 3, horizontal part of the duodenum up to the outlet of the superior mesenteric artery of the aorta (Figure 1). Mesenteric shrinkage was defined as a mesenteric soft tissue mass exhibiting typical characteristics of a local metastasis from the SI-NENs, which exerts a desmoplastic reaction and caused drawing in of the adjacent mesentery and small bowel loops, possibly encasing and occluding the mesenteric vessels and thereby causing a mixed arterial and venous ischemia of the affected bowel loops.

The proliferation rate was determined by the Ki-67 index and SI-NENs were classified according to the 8th edition of the UICC TNM classification [16,17]. Pathology reports were additionally re-reviewed for tumor location, number and size of tumors, and length of resected small bowel.

The patients were subjected to primary tumor resection with lymphadenectomy in the presence of bowel symptoms and also as a prophylactic after a multidisciplinary tumor board decision in cases of good physical condition. Liver metastases were synchronously removed when oncologically indicated and technically possible. For patients with diffuse metastatic SI-NENs without significant comorbidities, our multidisciplinary tumor board always recommends locoregional resection with at least 8 lymph nodes—independent of symptoms—to avoid local complications such as obstruction, bleeding and ischemia, which is in line with current German and ENETS guidelines [11,18]. In the case of level 3 (behind the pancreas) lymph node involvement on imaging in diffuse metastasized SI-NENs, an operation to solve the problem was only indicated if significant symptoms of bowel obstruction or ischemia were present. In the very rare cases of locally restricted SI-NENs with level 3 lymph node involvement, locoregional resection was attempted in otherwise fit patients, even if multivisceral resections, e.g., pancreatic head resection and vessel replacement, might be necessary.

Perioperative intravenous octreotide infusion (3 mg/24 h) was applied routinely to minimize the risk of carcinoid crisis during surgery.

### 2.2. Surgery

Patients were operated on with an extended abdominal midline incision. As an initial procedure, the intestine was explored from the ligament of Treitz to the ileocecal valve to identify the primary lesion, possible multiple SI-NENs and mesenteric lymph node metastases, as well as to anticipate the extent of small bowel resection. The right colon and small intestinal mesentery were always mobilized from posterior adhesions toward the retroperitoneum up to the level of the horizontal part of the duodenum and the lower border of the pancreatic body. Tumorous or fibrotic adhesions to the serosa of the horizontal duodenum frequently required sharp transection. The superior mesenteric artery and vein were identified first below the pancreatic body and then followed dorsally in the elevated mesenteric root. After this step, it was generally possible to determine the level of mesenteric metastatic extension along the major mesenteric vessels (Figure 1), which was used to judge whether the mesenteric metastases were completely resectable. Level 3 lymph node metastases are rarely completely resectable.

Conventional approach (Con-LA, pizza pie technique, Figure 2A and Figure 3A): The proximal and distal resection borders at the small bowel were determined and reined in. From the reined-in bowel edges, the mesenteric peritoneal surface was then superficially incised towards the most central lymph node metastasis along the superior mesenteric artery (SMA). Thereafter, the mesentery was divided with claps and sutures, or vessel sealing devices along these marked dissection lines. Finally, the bowel was transected and the specimen was removed.

Vessel-sparing approach (VS-LA, Figure 2B and Figure 3B): This approach has been routinely performed since 2018, mainly by one surgeon (D.K.B.). The AMS and VMS vessels were exposed by incision of the peritoneum ventrally over the mesenteric root. Lymphadenectomy always started at level II at the lower border of the pancreas body or at the horizontal part of the duodenum. Removal of the mesenteric soft tissue was accomplished by incising the peritoneum longitudinally over the mesenteric artery and by transecting fibrotic adhesions. Then, a retrograde dissection of the soft tissue along the artery and main mesenteric vein distally from the tumor capsule was performed with either bipolar forceps or bipolar scissors. It was always the goal to preserve the ileocolic vessels and important vascular collaterals and arcades along the intestine. Resection of the reined intestinal segment involved was postponed until dissection of the mesenteric tumor tissue was complete.

In both techniques, the bowel continuity was reconstructed in all cases with a sutured, double-layer side-to-side anastomosis. In cases of macroscopically questionable perfused small bowel ends, Indo-Cyanin-Green (ICG) near infrared angiography was used to determine the blood supply at the anastomotic site as described previously by our group [19].

Postoperative complications were classified according to Dindo-Clavien [20], and only clinically relevant complications > 2 were considered for analysis.

The patients were followed until death or the evaluation date 31 March 2022 at the Departments of Visceral, Thoracic and Vascular Surgery, or Gastroenterology and Endocrinology of the Marburg University hospital. Follow-up investigations comprised a physical examination, measurement of serum CgA, and cross-sectional imaging with MRI and/or functional somatostatin receptor imaging. Follow-up examinations took place every 6 months according to the ENETS guidelines [18,21]. Local SI-NEN recurrence was defined as either bowel recurrence or mesenteric lymph node metastases confirmed by functional SRS imaging and/or histology.

### 2.3. Statistics

Variables are presented as medians with ranges or means with SDs, or counts and percentages for categories as appropriate. Differences between groups were assessed using the Mann–Whitney test, χ^2^ tests and 2-sided *t* tests for unmatched data as appropriate. All tests were 2-sided unless stated otherwise. A value of *p*  < 0.05 was considered to be significant for all tests.

## 3. Results

Between February 2014 and December 2021, 102 patients had bowel resections for SI-NENs; of those, 48 were right hemicolectomies and 54 were small bowel resections only. Of the 54 patients with small bowel resections, 27 underwent VS-LA and the other 27 underwent Con-LA. In both groups, two patients each had bowel reoperations for local recurrences after surgery in other hospitals and were therefore excluded from analysis. One patient in the Con-LAD group underwent emergency surgery due to acute small bowel obstruction. The median age of patients at operation was 64 years in the VS-LA group and 61 years in the Con-LA group (*p* > 0.05). Symptoms of bowel obstruction and/or abdominal pain were present in 15 (60%) and 9 (36%) of the VS-LA and Con-LA groups (*p* = 0.09). The rate of diffuse metastatic disease was similar in the VS-LA and Con-LA groups (72% vs. 60%, *p* = 0.38). The prevalence of mesenteric shrinkage was similar between groups (84% vs. 64%), but the macroscopic involvement of lymph nodes at level 2 was higher in the VS-LA group (88%) compared to the Con-LA group (60%, *p* = 0.02, Table 1)

The prevalence of multiple tumors (32% vs. 36%, *p* = 0.77) as well as the median number of SI-NENs (3 vs. 4) was comparable. Pathological examination revealed that in the VS-LA group with median 40 (11–65) cm, significantly less small bowel was resected compared to the Con-LA group with median 65 (23–190) cm (*p* = 0.0007). The median number of harvested lymph nodes, however, was similar (13 vs. 13, *p* = 0.78). The rate of achieved local (mesenteric) R0 resection was also comparable (72% vs. 84%). In the VS-LAD group, where the lymph node metastasis and mesenteric tissue were removed from the vessels, there were slightly more R1 resections compared to the Con-LAD group, where lymph nodes, mesenteric tissue and vessels were resected. The lymph node ratio >0.2 tended to be higher in the VS-LA group (72%) compared to the Con-LA group 44% (*p* = 0.05). The pathological data are summarized in Table 2.

The median follow-up was significantly longer in the Con-LA group compared to the VS-LA group (63 vs. 24 months, *p* = 0.003). One (4%) patient in the VS-LA and seven (28%) patients in the Con-LA group experienced clinically relevant postoperative complications (*p* = 0.02). Six of seven clinically relevant complications in the Con-LAD group were directly related to the operation (intraabdominal hematoma (n = 3), bile leakage after simultaneously resected liver metastases (n = 1) and gastrointestinal hemorrhage from the anastomosis (n = 2)). One patient in each group suffered a pleural effusion and received a chest tube. More than 3 months postoperatively, 1 (4%) and 10 (40%) patients in the VS-LA and Con-LA groups, respectively, complained about intermittent abdominal pain (*p* = 0.002). Only 1 (4%) patient in the VS-LA and 3 (12%) in the Con-LA groups had symptoms of bowel obstruction (*p* = 0.29), whereas 1 of 8 (13%) stage II/III patients with VS-LA and 2 of 13 (15%) stage II/III patients with Con-LA and complete tumor resection experienced diarrhea (Table 3). None of the patients in either group experienced locoregional disease recurrence according to somatostatin-receptor imaging.

At the evaluation date, 5 of 24 (21%) patients in the VS-LA group had no evidence of disease, 19 (79%) were alive with disease and no patients had died of disease compared to 7 (29%), 13 (55%) and 2 (8%) patients in the Con-LA group (Table 3).

## 4. Discussion

To best of our knowledge, this is the first study comparing the results of VS-LA and Con-LA with regard to complications, radicality and postoperative function. Primary tumor resection plus lymphadenectomy is the treatment of choice for SI-NEN stage III and a good palliative option in symptomatic stage IV SI-NENs for resolving bowel obstruction and/or ischemia. Prophylactic primary tumor resection in non-resectable stage IV disease is controversial with inconclusive results in retrospective studies. However, in cases where surgery is indicated for SI-NENs, lymphadenectomy is an important surgical step and should be performed [11,12]. Here, we demonstrate that VS-LA is superior compared to Con-LA with regard to sparing a significant length of a resected small bowel (about 50%) and a lower rate of clinically relevant postoperative complications (4% vs. 28%). In addition, VS-LA tended to be associated with less abdominal pain and diarrhea after 3 months postoperatively in completely resected locoregional SI-NEN stages I–III. Moreover, the radicality appears to be equal with about median 13 resected lymph nodes, a similar local R0 status and no local recurrence during a follow-up of at least 2 years in both groups, although follow-up was significantly shorter in the VS-LA group.

Strategically retrograde VS-LA should be performed before the bowel resection [14,15] because the remaining vascularized intestine after mesenteric dissection will guide the extent of bowel resection for vascular purposes. In the case of a macroscopically questionable blood supply, NIR-angiography of the small bowel can be performed to clarify the blood perfusion [19,22].

Despite consensual agreement about the need for lymphadenectomy and its recommendation in previous ENETS guidelines [11,12], up to 20% of patients did not have any lymph node resection during SI-NEN surgery in some large, recently published series [23,24]. Lymphadenectomy may be challenging, especially in the presence of extensive mesenteric fibrosis and shrinkage, or large metastatic lymph nodes surrounding the superior mesenteric vessels. Retrospective registry analyses have suggested that at least eight (or possibly 12) removed lymph nodes are needed to improve overall survival [23,24,25]. Moreover, French guidelines have suggested discussing a ‘re-intervention’ after postoperative evaluation by F-DOPA-PET or 68Ga-PET, if fewer than eight lymph nodes have been resected [26]. This situation is most frequently observed after emergency surgery. The upper limit of lymphadenectomy is less consensual. In the absence of a retro-pancreatic target on preoperative imaging, lymphadenectomy is usually conducted along the trunk of the superior mesenteric vessels until the inferior body of the pancreatic body. However, Pasquer et al. reported the presence of lymph node skip metastases in 14 of 21 mainly metastatic patients and suggested that a systematic lymph node dissection up to the retropancreatic area should be realized [27]. The risk/benefit ratio of such a procedure in patients without liver metastases must be demonstrated because of the potential morbidity of such an extensive lymphadenectomy.

Lymphatic mapping may be an interesting way to determine the limits of the lymph node dissection using isosulfan/methylene blue, near-infrared fluorescent lymphangiography, but this practice is not yet performed as standard, nor is it strongly recommended by any guideline [11,12,18,28].

Bowel resection is generally not a key issue. The exact procedure depends on (i) the number of palpated primary tumors, (ii) their exact location generally in the last portion of the ileum and (iii) the remnant vascularized intestine after mesenteric dissection. For SI-NENs more than 30 cm proximal of the ileocecal valve, a small bowel resection is technically possible in most cases and the best option in order to keep in place the ileocecal valve and to potentially limit intestinal symptoms [11,12,18]. However, a right hemicolectomy is required in the majority of patients with tumors within 30 cm distance to the ileocecal valve, which is almost half of patients with SI-NENs [6]. This holds especially true in cases of a strong mesenteric shrinkage. Whatever the procedure, the maximum possible length of small bowel must be preserved to preclude malabsorption and diarrhea, especially bile-salt-induced diarrhea [29].

The present study has some limitations because of its retrospective design with the implicated bias. Moreover, the postoperative quality of life was not documented with validated standardized questionnaires. However, this is the first study analyzing the results of VS-LA and Con-LA in a well-defined, short-term collected cohort of SI-NEN patients with regard to complications, radicality and postoperative function.

## 5. Conclusions

In conclusion, VS-LA can be performed safely and should be considered for SI-NENs more than 30 cm proximal to the ileocecal valve sparing length of resected small bowel.

## Figures and Tables

**Figure 1 cancers-14-03610-f001:**
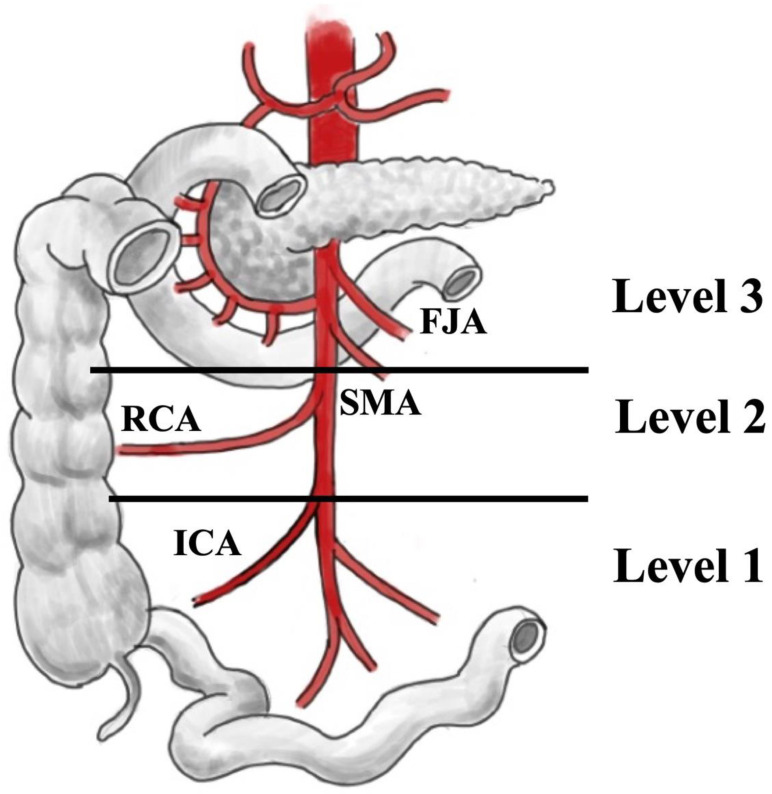
Levels of locoregional lymph node involvement in SI-NENs (modified based on Ohrvall et al. [7]). SMA—superior mesenteric artery; ICA—ileocolic artery; RCA—right colic artery; FJA—first jejunal arteries; D—duodenum.

**Figure 2 cancers-14-03610-f002:**
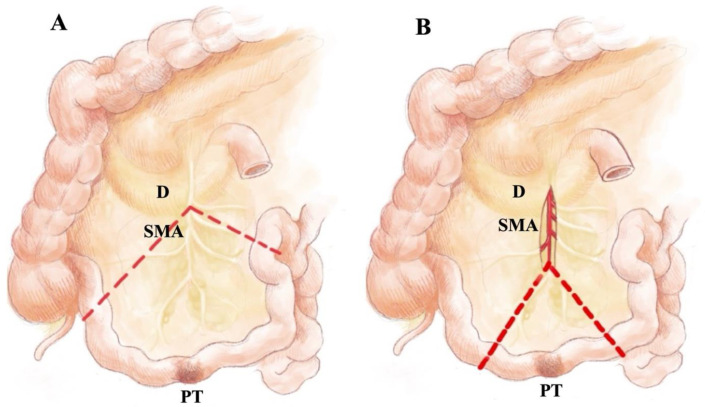
Schematic graph of conventional (**A**) and vessel-sparing lymphadenectomy (**B**) in SI- NENs. Red dashed line—dissection line; encircled area in B—superior mesenteric artery as well as right colic artery and vein were dissected free from soft tissue but not transected. D—duodenum; SMA—superior mesenteric artery; PT—primary tumor.

**Figure 3 cancers-14-03610-f003:**
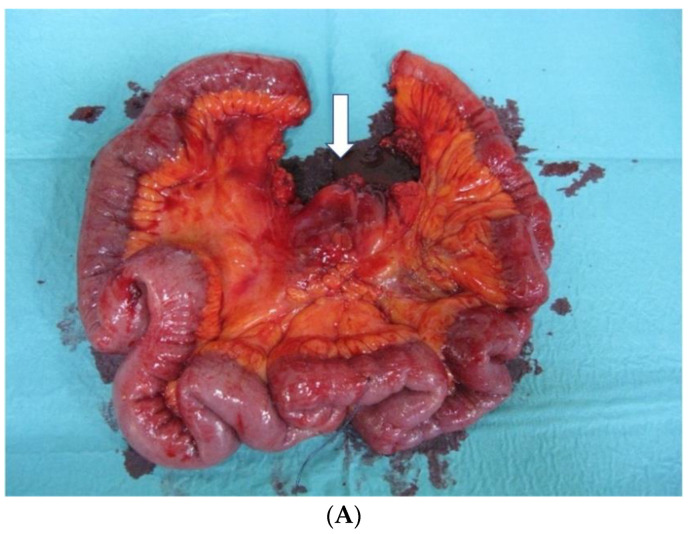
(**A**) Small bowel specimen after Con-LA: arrow—lymph node metastases level 2. (**B**) Situs after VS-LA. D—duodenum; ICA—ileocolic artery; SMA—superior mesenteric artery.

**Table 1 cancers-14-03610-t001:** Clinical characteristics and intraoperative findings in patients with SI-NENs undergoing small bowel resection.

Variable	VS-LA (*n* = 25)	Con-LA (*n* = 25)	*p*-Value
Age at surgery (median, years)	64 (28–82)	61 (38–80)	0.46
Female gender	8/25 (32.0%)	13/25 (52%)	0.16
Symptoms of bowel obstruction and/or abdominal pain	15/25 (60%)	9/25 (36%)	0.09
Diffuse metastatic disease	18/25 (73%)	15/25 (60%)	0.38
Mesenteric shrinkage	21/25 (84%)	16/25 (64%)	0.11
Macroscopic lymph node metastases ≥ level 2 ^#^	22/25 (88%)	15/25 (60%)	0.02

#—modified according to Ohrval et al. ([7], Figure 1).

**Table 2 cancers-14-03610-t002:** Pathological findings after small bowel resection for SI-NENs.

Parameter	VS-LA (*n* = 25)	Con-LA (*n* = 25)	*p*-Value
Multiple tumors	8/25 (32%)	9/25 (36%)	0.77
Median number of tumors in patients with multiple tumors	3 (2–10)	4 (2–14)	0.33
Mean distance largest tumor to cecal valve (cm)	50 (30–110)	75 (20–250)	0.04
Median length of resected small bowel (cm)	40 (11–65)	65 (23–190)	0.0007
Median number of resected lymph nodes	13 (4–58)	13 (2–51)	0.78
Lymph node ratio > 0.2	18/25 (72%)	11/25 (44%)	0.05
Local R0 resection	18/25 (72%)	21/25 (84%)	0.32
Tumor stage			
II	0	1 (4%)	0.32
III	8 (32%)	12 (48%)	0.15
IV	17 (68%)	12 (48%)	0.16

**Table 3 cancers-14-03610-t003:** Postoperative course and follow-up.

Parameter	VS-LA (*n* = 25)	Con-LA (*n* = 25)	*p*-Value
Median follow-up (months)	24 (3–91)	63 (6–94)	0.003
Postoperative complications > 2 Dindo–Clavien	1/25 (4%)	7/25 (28%)	0.02
Abdominal pain *	1/25 (4%)	10/25 (40%)	0.002
Diarrhea *			
All stages	5/24 (21%)	8/25 (32%)	0.34
Only stages I–III	1/8 (13%)	2/13 (15%)	0.31
Symptoms of bowel obstruction *	1/24 (4%)	3/25 (12%)	0.29
Disease status at last follow-up			
AWD	19/24 (79%)	13/24 (55%)	0.07
NED	5/24 (21%)	7/24 (29%)	0.52
DOD	0/24	2/24 (8%)	0.16
DURC	0/24	2/24 (8%)	0.16

*—after 3 months postoperatively, AWD—alive with disease, NED—no evidence of disease, DOD—dead of disease, DURC—dead of unrelated cause.

## Data Availability

The data presented in this study are available on request from the corresponding author.

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
