# Peer review of "Vessel-Sparing Lymphadenectomy Should Be Performed in Small Intestine Neuroendocrine Neoplasms"

_cancers, 2022, doi:10.3390/cancers14153610_

Round 1

Reviewer 1 Report

The authors provide a review of surgically treated patients for intestinal NET using two different surgical techniques. The paper could be interesting but needs some clarifications/corrections:

1.- "In both groups two patients each had bowel reoperations for local recurrences after surgery in other hospitals" => Should we understand that these surgeries were prior to the one analyzed in this study? In that case, it would be more appropriate to exclude these patients, since both the length of the bowel and the mesenteric approach could be conditioned by the previous surgery.

2.- There are some variables that would be of great interest but that the authors do not offer:

- How many patients underwent emergency surgery?

- What was the number of tumors in the intestine? Was it the same in both groups?

- What is the percentage of patients with metastatic nodes in each group?

- What were the reasons for the resections not considered as R0? Unresectable local disease or distant disease?

- The authors should indicate which were the clinically relevant postoperative complications.

3.- Due to the slow evolution of these tumors, it does not seem appropriate to include patients with few months of follow-up in the analysis of recurrences. Although the median follow-up is two years, there are patients with less than 3 months of follow-up. It would be more appropriate to include only patients with a longer follow-up, for example, only those operated until 2020, instead of 2021. It should also be indicated at what time the recurrences were diagnosed and the disease-free survival. Only by considering this and the average follow-up could an approximation of the oncological soundness of the procedure.

4.- "Here we demonstrate that VS-LA is superior compared to Con-LA with regard to sparing a significant length of resected small bowel (about 50%)" => The difference of the medians is 29 cm. This difference is likely to be of little clinical significance. The authors report that these patients have less diarrhea, but do not indicate how to assess it. All these limitations are intrinsic to the type of study, so we should be less categorical when indicating the superiority of the technique.

There are several typographical errors throughout the text and tables. The use of decimals should be uniform throughout the text. 

Author Response

Reviewer 1

1.- "In both groups two patients each had bowel reoperations for local recurrences after surgery in other hospitals" => Should we understand that these surgeries were prior to the one analyzed in this study? In that case, it would be more appropriate to exclude these patients, since both the length of the bowel and the mesenteric approach could be conditioned by the previous surgery.

Response: As suggested, we excluded two patients in each group who underwent reoperation. Accordingly, all statistical numbers were recalculated (Tables 1-3 and section “Results”).

2.- There are some variables that would be of great interest but that the authors do not offer:

- How many patients underwent emergency surgery?

Response: One patient of the Con-LAD group underwent emergency surgery, because of acute small bowel obstruction. We added this information in the section “Results”, page 12, line 6,7.

- What was the number of tumors in the intestine? Was it the same in both groups?

Response: As stated in Table 2 the number of patients with multiple tumors did not differ significantly between groups. As suggested, we added now also for both groups the median numbers of removed tumors in patients with multiple Si-NEN in Table 2: VS-LAD group 3(2-10), Con-LAD group 4(2-14). The numbers of resected tumors did not differ significantly between groups (p=0.33).

- What is the percentage of patients with metastatic nodes in each group?

Response: As already outlined in the result section (page 12, line 10,11) and table 2 100% of patients in the VS-LAD group and 96% in the Con-LAD group had metastatic lymph nodes. In addition, the lymph node ratio (metastatic lymph nodes/resected lymph nodes >0.2) was shown in Table 2.

- What were the reasons for the resections not considered as R0? Unresectable local disease or distant disease?

Response: As already outlined in table 2 the resection status refers to the local situation within the mesentery. We added this on page 13, line 1. In the VS-LAD group, where the lymph node metastasis and mesenteric tissue were removed from the vessels, resulted more R1 resections compared to the Con-LAD group, where lymph nodes, mesenteric tissue and vessels were resected. We pointed this now more out in the “Result” section (page 13, lines 2-4).

- The authors should indicate which were the clinically relevant postoperative complications.

Response: As suggested the clinically relevant postoperative complication were now explained in the results section (page 13, lines 10-13). Six of 7 clinically relevant complications in the Con-LAD group were directly related to the operation (intraabdominal hematomas, bile leakage after simultaneously resected liver metastases, gastrointestinal hemorrhage from the anastomosis). One patient in each group suffered of pleural effusion and received chest tubes.

3.- Due to the slow evolution of these tumors, it does not seem appropriate to include patients with few months of follow-up in the analysis of recurrences. Although the median follow-up is two years, there are patients with less than 3 months of follow-up. It would be more appropriate to include only patients with a longer follow-up, for example, only those operated until 2020, instead of 2021. It should also be indicated at what time the recurrences were diagnosed and the disease-free survival. Only by considering this and the average follow-up could an approximation of the oncological soundness of the procedure.

Response: We agree with the reviewer that a shorter follow-up in the VS-LAD group is a limitation. However, the only important thing regarding the reported patient cohort is local recurrence, since at not recurrence at all or even disease-free survival, since the 73% and 60% of patients had diffuse metastatic disease and were operated in palliative intention. As pointed out in the results and discussion section no local mesenteric recurrences were observed so far. Since the vessel-sparing lymphadenectomy was first introduced in 2018 and thereafter used as the preferred procedure we would lose 8 patients in the VS-LAD group from the analysis, if we would confine the cohort until 2020. This would make the analysis less meaningful and statistically questionable. Moreover, the major message of the presented data is that VS-LA can be safely performed, spares small bowel and might result in less postoperative short- and long-term complications. Therefore, we did not withdraw the patients from 2021. 

4.- "Here we demonstrate that VS-LA is superior compared to Con-LA with regard to sparing a significant length of resected small bowel (about 50%)" => The difference of the medians is 29 cm. This difference is likely to be of little clinical significance. The authors report that these patients have less diarrhea, but do not indicate how to assess it. All these limitations are intrinsic to the type of study, so we should be less categorical when indicating the superiority of the technique.

Response: We partially agree with the reviewer. Because of all the intrinsic limitations of our study we rephrased our conclusion in the abstract and the discussion section as follows: “VS-LA seems to be oncologically safe should be considered in small bowel resections for SI-NEN”. In addition, it might also be clinically significant to spare about 30cm of ileum with regard of resorption of vitamin B12 etc. As reported in the method sections the clinical parameters diarrhea and postoperative pain or obstruction were obtained by interviews of the patients during follow-up. It was also stated that no established questionnaire was used to document this information.       

There are several typographical errors throughout the text and tables. The use of decimals should be uniform throughout the text. 

Response: We apologize and corrected all spelling errors throughout the text.

Reviewer 2 Report

This was a very interesting retrospective analysis of a single center's experience with vessel sparing small bowel NET resection. The impact on QOL was notable when the two groups but I was left with a few questions. 

- Is there any information regarding the surgeons who participated? Was there a certain number of surgeons who favored one approach over another? Is there any information from tumor board consensus that guided the choice of surgery? 

- for patients with metastatic disease, what was the rationale for resection in the 63% (VS-LA) and 44% (Con-LA) to undergo resection of the primary? What was the breakdown for patients with metastatic disease that had symptoms prior to surgery? what was the breakdown of > level 2 lymph nodes for locally advanced versus metastatic disease?

- what is the breakdown for abdominal pain and bowel obstruction based on stage? 

Inclusion of this information would help provide a better sense of the practice patterns at your center and allow the reader to determine if it is translatable to their practice. 

Given the small numbers, the heterogeneity of the patient population, as well as the retrospective nature of the study, you may want to temper your conclusion and recommendation that this surgical technique be employed in all patients undergoing small bowel NET resection. 

Author Response

Reviewer 2

- Is there any information regarding the surgeons who participated? Was there a certain number of surgeons who favored one approach over another? Is there any information from tumor board consensus that guided the choice of surgery? 

Response: The vessel-sparing approach was performed mainly by one surgeon as stated in the method section (page 7, section “vessel-sparing approach”). He at least supervised all these procedures, most times he was the leading surgeon. The conventional approach was performed by the aforementioned surgeon and two others experienced surgeons. The type of procedure was not decided by the multidisciplinary tumor board, it was only proposed to avoid small bowel syndrome.

- for patients with metastatic disease, what was the rationale for resection in the 63% (VS-LA) and 44% (Con-LA) to undergo resection of the primary? What was the breakdown for patients with metastatic disease that had symptoms prior to surgery? what was the breakdown of > level 2 lymph nodes for locally advanced versus metastatic disease?

Response: We are grateful for these comments. For patients with diffuse metastatic SI-NEN without significant comorbidities our multidisciplinary tumor board always recommends locoregional resection with at least 8 lymph nodes – independent of symptoms - to avoid local complications like obstruction, bleeding and ischemia, which is line with current German and ENETS guidelines (Rinke et al. 2018, Niederle et al. 2016). In case of level 3 (behind the pancreas) lymph node involvement on imaging in diffuse metastasized SI-NEN an operation to solve somehow the problem was only indicated if significant symptoms of bowel obstruction or ischemia were present. In the very rare cases of locally restricted SI-NEN with level 3 lymph node involvement locoregional resection would be attempted in otherwise fit patients, even if multivisceral resections, e.g. pancreatic head resection and vessel replacement, might be necessary.  This information was now added in the method section (page 6, lines 9-18)      

- what is the breakdown for abdominal pain and bowel obstruction based on stage? 

Response:  9/25 patients in the Con-LAD group (5 stage IV, 4 stage III) and 15/25 patients in the VS-LAD group (11 stage IV, 4 stage III) suffered preoperatively from symptoms of bowel obstruction like pain. The level of locoregional lymph node involvement and its extent is decisive for abdominal symptoms. The breakdown based on stage does not provide any relevant information in the present cohort.

Inclusion of this information would help provide a better sense of the practice patterns at your center and allow the reader to determine, if it is translatable to their practice. 

Given the small numbers, the heterogeneity of the patient population, as well as the retrospective nature of the study, you may want to temper your conclusion and recommendation that this surgical technique be employed in all patients undergoing small bowel NET resection. 

Response: We agree with the reviewer and therefore changed our conclusion in the abstract and the discussion section as follows: “VS-LA seems to be oncologically safe should be considered in small bowel resections for SI-NEN.”

Reviewer 3 Report

In the present study, the author compared the results of PTR with retrograde vessel-sparing lymphadenectomy (VS-LA) to conventional lymphadenectomy (Con-LA).Results showed that VS-LA is superior compared to Con-LA with regard to complications, radicality and postoperative function. I only have one concern.

1.Please plot the survival or disease free survival curve along with Kaplan-Meier estimate for both VS-LA and Con-LA group.

Author Response

Reviewer 3

1.Please plot the survival or disease-free survival curve along with Kaplan-Meier estimate for both VS-LA and Con-LA group.

Response: 29 of 50 patients presented with stage diffuse IV disease, so a disease-free survival of the whole cohort cannot be calculated. To calculate the disease-fee survival of the remaining 8 patients in VS-LAD group and 13 patients in the Con-LAD group with locoregional disease (stage II/mostly III) is not meaningful, since none of the patients yet experienced recurrence and the median follow-up is significantly different. Instead, we calculated the overall survival, which is also limited by the different follow-up between groups.  The probability of 5-years survival is not statistically different between groups (p=0.20). If requested by the reviewer and the editor, the figure below can be incorporated in the manuscript
